# Prevalence of Attention Deficit Hyperactivity Disorder (ADHD) among Substance Use Disorder (SUD) Populations: Meta-Analysis

**DOI:** 10.3390/ijerph20021275

**Published:** 2023-01-10

**Authors:** Henrik Rohner, Nikolas Gaspar, Alexandra Philipsen, Marcel Schulze

**Affiliations:** Department of Psychiatry and Psychotherapy, Faculty of Medicine, University of Bonn, 53127 Bonn, Germany

**Keywords:** ADHD, SUD, prevalence, meta-analysis, opioid use disorder, cocaine use disorder, alcohol use disorder, addiction

## Abstract

(1) Background: Attention deficit hyperactivity disorder (ADHD) is characterized by a persistent pattern of age-inappropriate levels of inattention and/or hyperactivity/impulsivity that results in functional impairment at work, education, or hobbies and affects family life, social contacts, and self-confidence. ADHD is a comorbid condition associated with a prognosis of severe substance use disorder (SUD) and the early onset of such. The aim of this meta-analysis was to obtain the best estimate of the prevalence of ADHD in SUD populations. (2) Methods: A literature research was conducted using PUBMED^®^ and Web of Science^®^. The following search terms were used: [ADHD], [prevalence], and [substance use disorder]. RStudio^®^ was used for meta-analysis methods. (3) Results: In total, 31 studies were included. We estimate the prevalence of ADHD among SUD patients at 21%.

## 1. Introduction

Addiction has a huge impact in healthcare systems worldwide. One out of nine hospitalizations in the United States of America are made by patients with substance use disorder (SUD). Unfortunately, numbers have been increasing in recent years [1]. Furthermore, approximately 28% of all SUD patients die within fifteen years after seeking treatment, with a mean age under 50 years [2]. There are many different substances with addictive potential, e.g., alcohol, cocaine, or heroin [3].

The course of SUD is strongly influenced by social environment and existing comorbidities—both somatic and psychiatric. ADHD is a common comorbidity in SUD patients [4] with a more severe course of substance abuse [5]. ADHD is characterized by a persistent pattern of age-inappropriate levels of inattention and/or hyperactivity/impulsivity that results in functional impairment at work, education, or hobbies and affects family life, social contacts, and self-confidence [6]. Usually, ADHD symptoms become apparent in childhood. However, around 40–50% of this population continues to experience ADHD symptoms in adolescence and adulthood, while hyperactivity often diminishes and inattention stays more prevalent [7,8]. Research has shown that ADHD has a prevalence of 2.5% in adults and 3.4% in childhood [9,10]. Moreover, a meta-regression analysis showed that the geographical location and year of study were not associated with variability in ADHD prevalence estimates [9].

Patients with ADHD are more likely to develop SUDs [11] at a younger age [12]. Furthermore, a large population-based epidemiological study showed that ADHD symptoms were associated with significantly increased risks for alcohol use disorder, illicit drug use, and multiple substance use [13]. However, the reason for the increased association between ADHD and SUD is unknown, although some authors suggest that substance abuse represents an attempt to self-medicate ADHD symptoms [14]. Thus, therapeutic strategies of both disorders should be taken into consideration while treating young adults with SUD and ADHD. Psychopharmacological treatment alone does not appear to be particularly effective at treating SUD in currently active substance-using patients with ADHD. Multimodal therapies may be effective at treating patients with ADHD and comorbid SUD [15].

Current data indicate an ADHD prevalence of 21.5% in SUD populations [16]. However, there were many attempts to estimate the prevalence of ADHD among SUD populations over the last decades, with inconsistent data ranging from 5.22% [17] to 62% [18]. Investigations of different SUD populations being addicted to different substances show that an increased prevalence of ADHD can be found in almost every SUD population. Both stimulant substances and sedative substances appear to be abused by ADHD patients frequently. There are studies in populations of cocaine users showing ADHD prevalence between 14.5% [19] and 20.5% [20]. In populations of patients with alcohol addiction studies, there was an estimated ADHD prevalence between 7.7% [21] and 62% [18], and among opioid users, an ADHD prevalence of 16.8% was shown [22]. The most current meta-analysis we know of was published in 2012. Here, the estimated prevalence of ADHD among SUD populations was 23.1%. Furthermore, the meta-analysis showed that ADHD prevalence in adolescents was 25.3% and that ADHD prevalence in adults was 21.0% [23]. However, up to this day, there is no screening for ADHD in daily clinical routine treatment of patients with SUD. We therefore decided to perform a new meta-analysis including the most recent studies in order to highlight the impact of ADHD in SUD more profoundly.

Objectives: The aim of this meta-analysis is to obtain an estimate of ADHD prevalence among adult SUD patients and SUD subpopulations for different substances, which could lead to a better understanding of the correlation between these two different psychiatric diseases, thus allowing the potential necessity of implementing screening tools for ADHD in SUD and vice versa to be studied.

## 2. Materials and Methods

### 2.1. Search Strategies

The structure of the meta-analysis was based on the preferred reporting items for systematic reviews and meta-analyses (PRISMA) 2020 guideline [24]. Literature research of the PUBMED^®^ and Web of Science^®^ databases took place using the search terms [ADHD], [prevalence], and [substance use disorder]. The time period for inclusion was from 1970 to 2022. Reference lists of obtained articles were also considered. The search was conducted from 11 May 2022 to 7 July 2022.

### 2.2. Study Selection

The following inclusion criteria were used (1) a publication of the research paper in a peer-reviewed journal, (2) the formal diagnosis of SUD according to DSM [25] or ICD [26], (3) the formal diagnosis of ADHD in adults or adolescents according to DSM [25] or ICD [26] (diagnosis based only on a self-report questionnaire is not sufficient), and (4) whether systematic and sufficient screening among SUD populations had been performed. The following exclusion criteria were used (1) publications only using secondary analysis of data (e.g., systematic reviews), (2) the paper not being eligible in English or German language, and (3) a lack of information necessary for the meta-analysis.

For an overview of study inclusion and exclusion, see Figure 1. We assessed the risk of bias based on seven domains and used risk-of-bias VISualization (robvis) to create risk-of-bias plots for the included studies [27].

### 2.3. Recorded Variables

The extracted variables from each study were sample size, gender distribution, the mean age of the participants, the prevalence of ADHD among the population, and the main substance of abuse. For the purpose of this meta-analysis, we only extracted subpopulations of opioid-, cocaine-, and alcohol users. Studies with other substances as the main substance or with missing information on a main substance were declared to be various types of substance abuse. If sub-populations were specified within a study, we tried to take them into account in the analysis. Data were extracted and checked from each study by one and the same researcher (N.G.).

### 2.4. Meta-Analytic Approach

The meta-analytic procedure was realized using R-software library package metafor [version 2.0-0] [28]. A random-effects model was calculated based on logit transformation of single proportions to obtain overall proportion. Inverse variance weighting was chosen, and between-study variance was calculated using Der Simonian Laird estimator [29]. Heterogeneities were assessed with *Q* and *I*^2^ statistics. Conventions were followed by the interpretation of *I*^2^: values of 0.25, 0.50, and 0.75 correspond to low, moderate, and high between-trial heterogeneities [30]. The publication bias was assessed with funnel plots and an egger intercept. Further, since age was not further specified as an inclusion criterion, meta-regression with age as a moderator variable was performed.

We conducted the same methods on the three SUD subpopulations: cocaine, opioids, and alcohol.

## 3. Results

### 3.1. Included Studies and Sample Characteristics

We screened 1691 records and included 31 studies with a total participant size of 12,524. Three studies with opioids (*n* = 2357), seven studies with cocaine (*n* = 2974), and seven studies with alcohol (*n* = 2143) as the main substance of abuse were included. For a list of included studies, see Table 1. The risk of bias of the included studies was estimated as overall low see Figure 2 and Figure 3.

### 3.2. Meta-Analytic Findings

With a random-effects model, we determined the general ADHD prevalence among SUD patients of 21% (95% CI = [0.1741; 0.2548]). Significant heterogeneity was present (*I*^2^ = 95.8%, *Q* = 721.09; *df* = 30; *p* < 0.0001). A moderator analysis was conducted for age, suggesting that age has only a slight influence on heterogeneity, *Q* (*df* = 25) = 20.2909, *p* = 0.7314. For the Forest plot of the effect sizes and 95% confidence intervals, see Figure 4. The visual inspection of the funnel plots and the egger intercept (*z* = −1.69, *p* = 0.30) suggest the absence of a publication bias (see Figure 5).

While we were able to calculate the general prevalence of ADHD among SUD patients, there were unfortunately very few heterogenous data for the subpopulations for each substance on its own. Hence, the following data should be interpreted with caution.

For the subpopulation of cocaine users, we calculated an ADHD prevalence of 19% (95% CI = [0.1058; 0.3102]) with significant heterogeneity (*I*^2^= 96.7%, *Q* = 238.89; *df* = 8; *p* < 0.0001). For Forest and Funnel plots, see Figure 6.

For the subpopulation of opioid users, we calculated ADHD prevalence of 18% (95% CI = [0.0784; 0.3505]) with significant heterogeneity (*I*^2^= 97.4%, *Q* = 153.26; *df* = 4; *p* < 0.0001). For Forest and Funnel plots, see Figure 7.

For the subpopulation of alcoholics, we calculated ADHD prevalence of 25% (95% CI = [0.1845; 0.3360]) with significant heterogeneity (*I*^2^ = 92.5%, *Q* = 106.55; *df* = 8; *p* < 0.0001). For Forest and Funnel plots, see Figure 8.

Unfortunately, we were not able to conduct the moderator analysis on these subpopulations because of the small *n* among them.

## 4. Discussion

Our main aim was to obtain an estimate of the overall prevalence of ADHD in adult and adolescent SUD patients. We estimate the prevalence of ADHD among SUD populations at 21%. The estimated prevalence and substantial heterogeneity are consistent with the results of a former meta-analysis from 2012 [23] using similar inclusion and exclusion criteria, but our analysis included more studies and a different meta-analytic approach. Thus, approximately one out of five patients with SUD also suffer from comorbid ADHD in adulthood and adolescents.

By addressing individual SUD populations, we calculated the prevalence of 19% for the cocaine SUD population, 18% for the opioid SUD population, and 25% for the alcohol SUD population. These results suggest that the high prevalence of ADHD in adulthood can be found in SUD populations of all kinds of different substances of abuse. However, these results should be interpreted with caution since the number of included studies is very low and due to the presence of high heterogeneity in all models.

There are various potential reasons for the substantial heterogeneity in our analysis. One might think about local differences in substance abuse in ADHD appearance as a reason for this, depending on where the studies were conducted. Other explanations for the heterogeneity might be rater bias, the use of different screening and diagnostic tools, or the determination of different thresholds for the latter. Moreover, diagnosing SUD patients with ADHD is very difficult without intraindividual attention and adjustment to the patients and the in- or outpatient setting.

Moreover, there are many possible screening tools for ADHD among SUD patients that could be considered for daily psychiatric practice. One way to undertake ADHD screening could be the combination of the Wender Utah Rating Scale (WURS-k) [53] for symptoms in childhood and the German self-rating behavior questionnaire (ADHD-SR) [54] for symptoms in adulthood; combined, they have sensitivity of 94% and specificity of 56% [55]. Another method of screening could be the adult ADHD self-report scale v1.1 [56] alone with a sensitivity of 100%, a specificity of 71%, a positive predictive value of 0.52, and a negative predictive value of 1.0 [57]. Thus, future studies should investigate which screening tools and which diagnostic procedures are most appropriate and workable for SUD patients in daily practice. Ideally, this should be investigated in a multicentric study design and among different SUD subpopulations, documenting the different substances as specific as possible. Additionally, further scientific efforts should endeavor to determine which therapy concepts are best suited for patients with ADHD and SUD. Established psychological interventions for SUD could be an effective treatment for patients with SUD and ADHD [58].

Limitations: This meta-analysis suggests that the available data of ADHD in adulthood among SUD populations is very heterogeneous; hence, the current results should be interpreted very cautiously. Furthermore, we were not able to conduct the moderator analysis with the moderator age on the three subpopulations because of the small *n* among them. For accuracy reasons, we excluded 46 studies only using self-reported questionnaires for confirmation of the diagnosis of ADHD.

## 5. Conclusions

In today’s psychiatric clinical practice, ADHD in adulthood remains a highly underrated condition, especially when focusing on people suffering from SUD. This meta-analysis shows that every fifth patient suffering from SUD could be diagnosed with a comorbid ADHD if evaluated precisely for research purposes. In order to improve mental health care for this population, new concepts for diagnosis and treatment in daily psychiatric practice in in- and outpatient care must urgently be developed. If diagnosed earlier with ADHD, it is probable that the severity of the course of SUD could be attenuated or even the occurrence itself could be prevented. However, the important question of whether people suffering from ADHD are particularly more susceptible to develop SUD to certain substances unfortunately remains unanswered. Hence, future research efforts should attempt to investigate the prevalence of ADHD among SUD populations for specific substances or substance groups more intensely to address diagnostic and therapeutic means more individually in the future.

## Figures and Tables

**Figure 1 ijerph-20-01275-f001:**
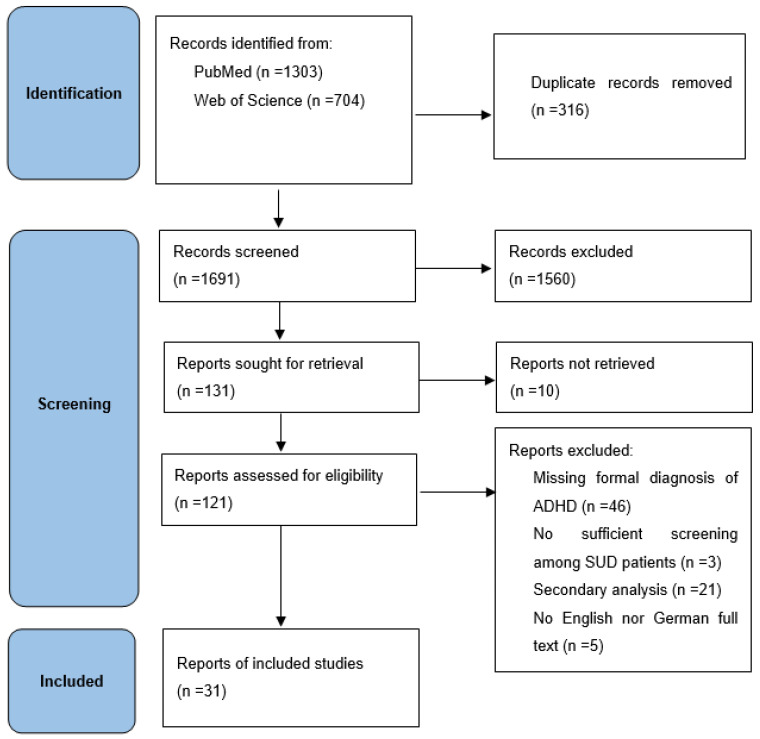
PRISMA 2020 [24] flow diagram.

**Figure 2 ijerph-20-01275-f002:**
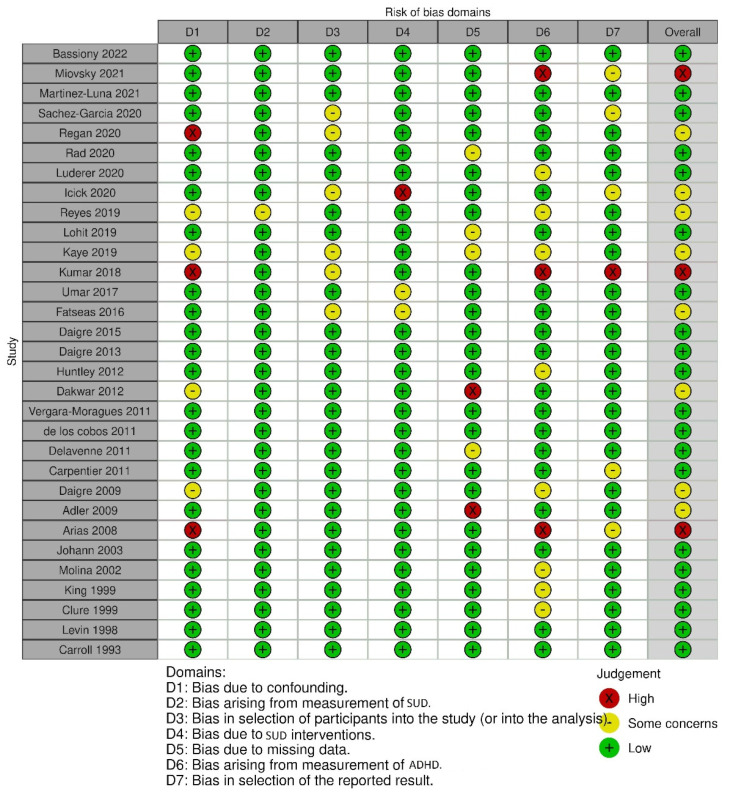
Risk-of-bias traffic light plot [27] of included studies.

**Figure 3 ijerph-20-01275-f003:**
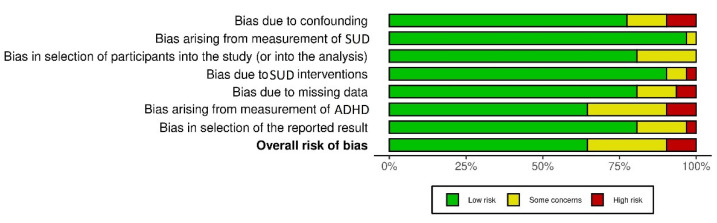
Risk-of-bias summary plot [27] of included studies.

**Figure 4 ijerph-20-01275-f004:**
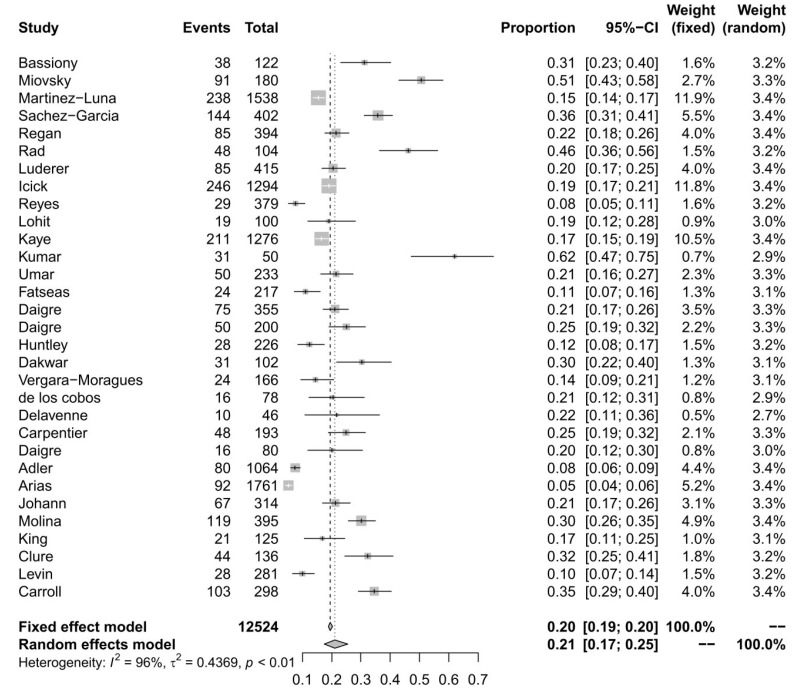
Forest plot of the effect sizes and 95% confidence intervals for ADHD prevalence.

**Figure 5 ijerph-20-01275-f005:**
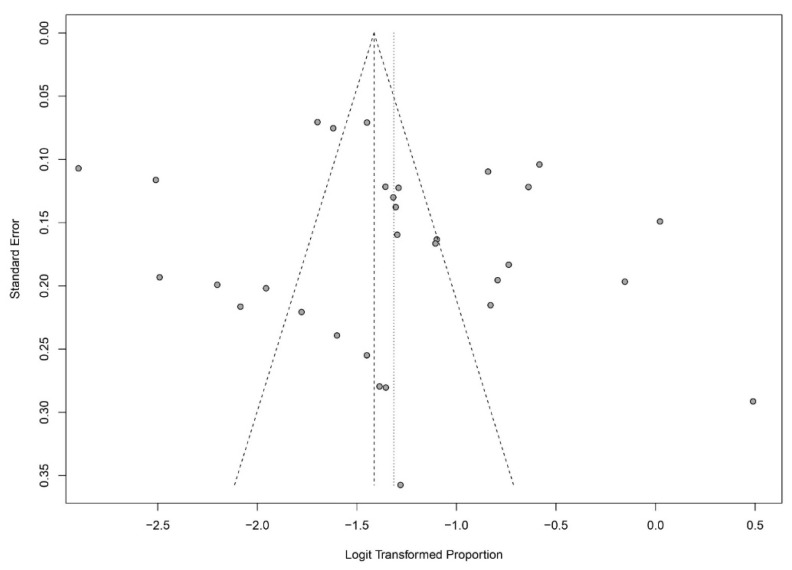
Assessment of publication bias with funnel plots for ADHD prevalence.

**Figure 6 ijerph-20-01275-f006:**
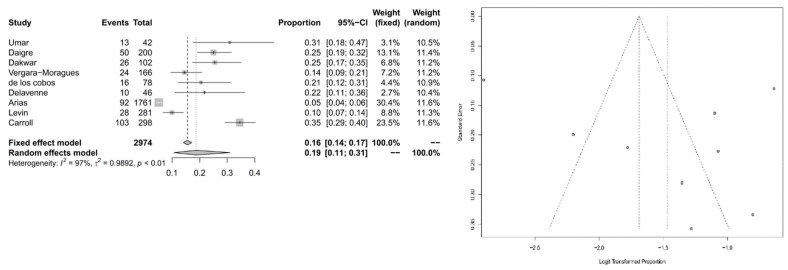
Forest plot of the effect sizes and 95% confidence intervals, and assessment of publication bias with funnel plots for the subpopulation of cocaine users.

**Figure 7 ijerph-20-01275-f007:**
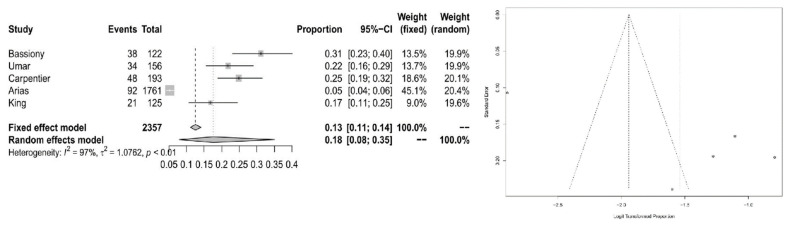
Forest plot of the effect sizes and 95% confidence intervals, and assessment of publication bias with funnel plots for the subpopulation of opioid users.

**Figure 8 ijerph-20-01275-f008:**
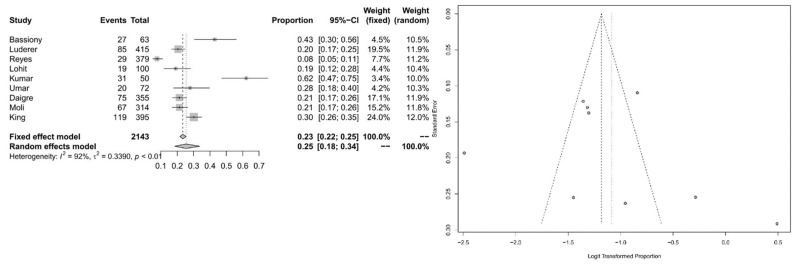
Forest plot of the effect sizes and 95% confidence intervals, and assessment of publication bias with funnel plots for the subpopulation of alcoholics.

**Table 1 ijerph-20-01275-t001:** Included studies.

Author	Year	Sex (male)	Mean Age	Age SD	ADHD Prevalence	*n*	Substance	Country of Origin
Bassiony et al. [12]	2022	83.6%	31.28	7.24	31%	122	opioid	Egypt
Miovsky et al. [31]	2021	76.7%	28.1	6.11	50.6%	180	various	Czech Republic
Martinez-Luna et al. [32]	2021	80%	32.9	10	15.5%	1538	various	Spain
Sanchez-Garcia et al. [33]	2020	79.6%	36.91	11.87	35.75%	402	various	International
Regan et al. [16]	2020	71.1%	16.33	1.15	21.5%	394	various	U.S.A.
Rad et al. [34]	2020	60.58%	na	na	46%	104	various	Romania
Luderer et al. [35]	2020	72.02%	45.35	10.2	20.5%	415	alcohol	Germany
Icick et al. [36]	2020	74%	40	11	19%	1294	various	International
Reyes et al. [21]	2019	65.4%	41.9	11.7	7.7%	379	alcohol	International
Lohit et al. [37]	2019	100%	40.68	na	19%	100	alcohol	India
Kaye et al. [38]	2019	na	na	na	16.53%	1276	various	International
Kumar et al. [18]	2018	100%	32.06	7.22	62%	50	alcohol	India
Umar et al. [39]	2017	82.8%	26.31	6.53	21.5%	233	various	Nigeria
Fatseas et al. [40]	2016	66.4%	37.7	10.6	11.1%	217	various	France
Daigre et al. [41]	2015	78.3%	36.15	10.43	21.12%	355	alcohol	Spain
Daigre et al. [4]	2013	87%	33.28	7.4	25%	200	cocaine	Spain
Huntley et al. [42]	2012	76.5%	39	10.3	12.2%	226	various	United Kingdom
Dakwar et al. [43]	2012	na	na	na	25	102	cocaine	U.S.A.
Vergara-Moragues et al. [19]	2011	91%	34.84	7.4	14.5%	166	cocaine	Spain
De los cobos et al. [20]	2011	81%	32.2	7.3	20.5%	78	cocaine	Spain
Delavenne et al. [44]	2011	95.65%	na	na	21.7%	46	cocaine	France
Carpentier et al. [45]	2011	83.42%	40.59	6.84	25.9%	193	opioid	Netherlands
Daigre et al. [46]	2009	80%	36.15	10.43	20%	80	various	Spain
Adler et al. [47]	2009	na	na	na	7.5%	1064	various	U.S.A.
Arias et al. [17]	2008	51.9%	38.37	7.67	5.22%	1761	various	U.S.A.
Johann et al. [48]	2003	83%	43.1	8.77	21.3%	314	alcohol	Germany
Molina et al. [49]	2002	63%	16.75	1.22	30%	395	alcohol	U.S.A.
King et al. [22]	1999	46%	37	7.75	16.8%	125	opiod	U.S.A.
Clure et al. [50]	1999	75.59%	34.3	0.78	32%	136	various	U.S.A.
Levin et al. [51]	1998	82%	33.7	0.4	10%	281	cocaine	U.S.A.
Carroll et al. [52]	1993	69%	27.7	6.06	34.6%	298	cocaine	U.S.A.

na = data not available.

## Data Availability

The data that support the findings of this study are available from the corresponding author upon reasonable request.

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
