# Peer review of "Prevalence of Attention Deficit Hyperactivity Disorder (ADHD) among Substance Use Disorder (SUD) Populations: Meta-Analysis"

_ijerph, 2023, doi:10.3390/ijerph20021275_

Round 1
Reviewer 1 Report
To conduct the meta-analysis, titled “Prevalence of attention deficit hyperactivity disorder (ADHD) 2 among substance use disorder (SUD) populations: Meta-analysis”, the authors used two databases in their search. I would like to appreciate the authors for this contribution. The results of the investigation are promising. For more specific recommendations on how to make the text simpler to comprehend, please see the comments below.
Abstract
Page 1, line 10 – The authors must reframe the initial sentence. It is illogical to describe ADHD as a disorder that co-occurs with SUD; instead, authors should describe the condition first before discussing the comorbidity.
The authors have included two databases, Pubmed and Web of Science. However, it is advised to include other databases (e.g., Embase, Medline) to ensure the review is thorough and the evidence is not missed.
Introduction
Technical and grammatical errors should be checked for in the article (page 1, lines 35, 40, and 42, among others).
“Investigations of different SUD populations …... prevalence of ADHD can be found in almost every SUD population”, It would be good to mention the populations the authors are talking about. Please provide the references.
Objectives should be explicitly stated.
Methods
What are the countries of origin of the included papers? What is the time period for inclusion?
Authors are advised to present the full search strategies for all databases, registers, and websites searched in the method section (refer to latest PRISMA guidelines -2020).
“Data were extracted ….. by one and the same researcher”. For systematic reviews and meta-analyses, it is advised that two researchers independently work on article screening and information retrieval and then arrive at a consensus.
How was the risk of bias calculated? Specify the tool used and present the findings.
Results and Discussions
The authors have discussed heterogeneity in results depending on where the studies were conducted. Authors can think about presenting a geographical location-specific prevalence of ADHD. For instance, there will be obvious differences in the prevalence based on country of study, ethnicity, etc.
Reviewer 2 Report
The authors provided an important update regarding the prevalence of ADHD in SUD populations. However, as the authors have discussed in the manuscript, the data for different SUD subpopulations are inadequate, which impedes drawing solid conclusions. In this case, I suggest adding more key words when search literatures, for example, “alcohol use disorder”, “opioid use disorder”.
Besides above major point, there are a few minor points.
1. Please use decimal point, rather than comma symbol, to indicate non-integers.
2. The figure legends of Fig 2, 3, 4, 5 are too simplify, please describe more details.
Reviewer 3 Report
Thank you for the opportunity to review "Prevalence of attention deficit hyperactivity disorder (ADHD) 2 among substance use disorder (SUD) populations: Meta-analysis." As a clinician and a research who works with people affected by SUD and ADHD I appreciated the goal of the study as well as the study design and methods.
There were a few areas that could be improved - namely, the introduction and discussion were rather brief, and additional literature, theory, and background information about assessment and treatment of comorbid substance use and ADHD could be cited. For example, see:
Zulauf CA, Sprich SE, Safren SA, Wilens TE. The complicated relationship between attention deficit/hyperactivity disorder and substance use disorders. Curr Psychiatry Rep. 2014 Mar;16(3):436. doi: 10.1007/s11920-013-0436-6. PMID: 24526271; PMCID: PMC4414493.
Mariani JJ, Levin FR. Treatment strategies for co-occurring ADHD and substance use disorders. Am J Addict. 2007;16 Suppl 1(Suppl 1):45-54; quiz 55-6. doi: 10.1080/10550490601082783. PMID: 17453606; PMCID: PMC2676785.
Young S, Woodhouse E. Assessment and treatment of substance use in adults with ADHD: a psychological approach. J Neural Transm (Vienna). 2021 Jul;128(7):1099-1108. doi: 10.1007/s00702-020-02277-w. Epub 2020 Nov 19. PMID: 33211196.
Additionally, the results are presented in a clear way but there is not enough integration of these findings with the existing literature within the discussion. It would be helpful to have authors provide more "interpretation" as to how their findings contribute to the existing literature/move the field forward.
Author Response
Please see the attachement.
